# LLM-Cite: Cheap Fact Verification with Attribution via URL Generation

## Abstract

Hallucinations are one of the main issues with Large Language Models (LLMs). This has led to increased interest in automated ways to verify the factuality of LLMs' responses. Existing methods either rely on: (a) search over a knowledge base (KB), which is costly especially if the KB must be updated frequently to keep up with fresh content, (b) LLM's parametric knowledge to fact-check claims, which is cheaper but does not give attribution and is limited to verifying claims related to knowledge acquired during pretraining. In this work, we present LLM-Cite, a cheap and easy to implement method that does not rely on any external search system while still providing attribution and the ability to verify fresh claims. Our key insight is to leverage an LLM to directly generate potential citation URLs for a given claim, and then use entailment checks to verify the claim against content of the URLs (which are fetched on-the-fly). We benchmark LLM-Cite on three datasets containing fresh and non-fresh claims generated by humans and models. We show that LLM-Cite performs comparable or better than existing methods on all categories of claims — importantly, without sacrificing attribution, or requiring costly external search — overall LLM-Cite is more than 45x cheaper than a Google Search based approach.

## 1 Introduction

Large Language Models (LLMs) have demonstrated tremendous progress in language understanding, and have wide ranging applications from summarization (Zhang et al., 2024) to question answering in domains like law (Yu et al., 2022) and medicine (Singhal et al., 2023). Nevertheless, LLMs are known to hallucinate and generate plausible sounding yet factually incorrect text (Maynez et al., 2020; Ji et al., 2022). Thus, automated methods to check the factuality of LLMs' responses is an important problem (Guo et al., 2022; Kenthapadi et al., 2024).

Existing methods to do automated fact-checking either (a) rely on an external index such as retrieval-augmented verification (Lewis et al., 2020) or LLMs with web search (Chern et al., 2023); (b) rely on an LLM's parametric knowledge to do fact-checking (Lee et al., 2021; Kadavath et al., 2022). Methods relying on an external index include FactScore (Min et al., 2023) where each atomic claim in an LLM's response is verified by doing an embedding-based retrieval against a static index (e.g. created from Wikipedia) and then using an LLM to verify if the claim is supported by the retrieved documents. One of the main limitations of such an approach is that it relies on a static corpus and thus cannot verify *fresh claims*—claims whose validity depends on the changing world, e.g., winner of the November 2024 election in the USA. Methods relying on external tools like Google Search can work with fresh claims but are quite expensive due to the high cost of Google Search API. Approaches relying on an LLM's parametric knowledge work by directly prompting the LLM to verify facts (Kadavath et al., 2022). They are straightforward and cheap but do not provide any attribution, and are also limited by the model's knowledge cutoff in verifying fresh claims. Table 1 includes a detailed comparison across existing methods.

Motivated by this, we tackle the following research question — *Can we design a simple, cheap method for fact verification with attribution that can also verify fresh claims?*

It is clear that to verify fresh claims, one has to rely on an external knowledge base for up-to-date information. This process typically involves (1) searching and retrieving potentially corroborating documents, and (2) checking whether the claim is entailed by any of those documents. In this

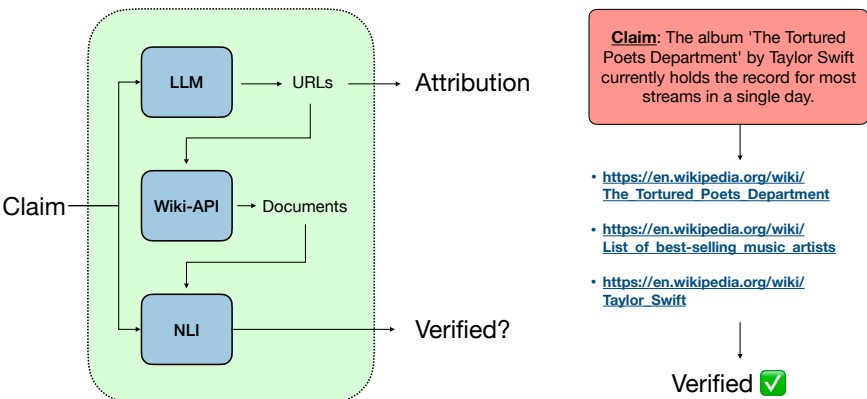

Figure 1: Illustration of LLM-CITE: (left) We use LLMs to generate candidate citation URLs, fetch the documents on-the-fly, and use a NLI model to check if claim is entailed by the documents ; (right) For a claim outside the knowledge cutoff of the LLM, the LLM can still generate valid and useful candidate URLs which can help verify the claim without doing web search.

work, we ask the radical question: *Can we off-load the expensive task of searching for potentially corroborating documents to an LLM?* At first, this seem counter-intuitive — if the LLM does not have up-to-date information about the knowledge base, how can we expect it to search through it? To our surprise, it turns out that LLMs can perform this task well for web domains like Wikipedia.

Our key insight is that the search can be made feasible by asking an LLM to identify **URLs** of top-k (potentially) corroborating documents. We call these *citation URLs*. The reason why this works is twofold. First, popular datasets used for pre-training LLMs (Gao et al., 2020; Raffel et al., 2019) include several URLs. During pre-training, the LLM memorizes many such URLs, and their association with the corresponding document contents. Our "search" prompt tries to elicit this information from the LLM. Second, URLs have a semantic structure and they typically include terms that are meaningfully tied to the document contents. Consequently, even if the URL is not seen by the LLM during pre-training, it can learn to guess its form based on the claim. For instance, the first URL mentioned in Figure 1 (right) is not seen in pretraining (since it was created after the knowledge cutoff), yet the LLM was able to generate it for the given claim.

Once we have identified candidate citation URLs, the subsequent steps are to fetch the most recent contents of these URLs, and check if they entail the claim. Notice that such content fetching, which can be automated with tools like `curl`, `wget`, etc., is far cheaper than a full web or domain search.

In summary, LLM-CITE consists of three steps: (1) use few-shot prompting with LLMs to generate multiple candidate citation URLs for a given claim ; (2) on-the-fly fetching of the fresh contents of the candidate documents ; (3) using a Natural Language Inference (NLI) model to verify if the claim is entailed by the fetched documents. Figure 1 shows an overview of our method. Overall LLM-CITE is very easy to implement, cheap, and can verify fresh claims; Table 1 show detailed comparison between LLM-CITE and existing fact verification methods.

We benchmark LLM-CITE on the task of fact-checking claims against Wikipedia.[1] We benchmark LLM-CITE against several leading fact verification methods, including, verification based on Google search. We evaluate on a variety of claims from three datasets, including, claims written by humans, claims from model generated responses, and fresh claims. Across all settings, we find that LLM-CITE is at par or better than the leading fact verification methods.

Our main contributions can be summarized as follows:

- We propose a very simple and cheap method, LLM-CITE, that can verify claims without requiring search through an external corpus.

---

[1]LLM-CITE can work with any popular web domain. We chose Wikipedia since it is widely accepted as truthful and unbiased, and due to the availability of public datasets of claims that can be verified against it.

- Across diverse model generated and human written claims, we empirically demonstrate that LLM-CITE can perform comparable or better than existing fact verification methods.

- We demonstrate that LLM-CITE can also verify fresh claims, which is lacking in static methods and is more than 45x cheaper then methods relying on Google Search.

## 2 LLM-CITE

LLM-CITE is a simple, cheap method that can verify any claim (both fresh and non-fresh) without requiring an external search index. It consists of three steps: (1) generating candidate citation URLs using LLMs (Section 2.1), (2) on-the-fly fetching of the documents (Section 2.2), (3) verifying the claim based on the fetched documents (Section 2.3). Step (1) leverages LLMs to bypass the expensive step of "searching" for corroborating evidence in traditional fact verification systems. Step (2) ensures that LLM-CITE makes judgments based on latest information, and therefore can verify fresh claims.

### 2.1 GENERATING URLS

Given a claim, the first step in LLM-CITE is to generate candidate citation URLs that may help in verifying the claim. We do this via prompting an LLM. Our prompts, shown in A.1 (Table 4 and 5), use few-shot examples along with instructions that ask an LLM to generate URLs of pages that can help verify the given claim.

While our prompt is straightforward, it is surprisingly effective and can work with any off-the-shelf LLM. This effectiveness stems from two properties of LLMs. First, URLs of popular web pages (e.g., from Wikipedia ) would be seen by the LLM along with the page content during pretraining. This allows the LLM to learn an association betwen the URL and the page content. Our prompt simply "elicits" this information from the LLM's parametric memory. Second, URLs typically have a semantic structure along with interpretable domain, sub-domain, and path names. This allows the LLM to effectively guess the URL even if it has not seen the claim or URL during pretraining; see 1st URL in Figure 1 for an example.

In order to account for errors during URL generation, we generate up to $m$ diverse URLs. A claim is verified if the contents of any one of the generated URLs corroborate the claim. Generating multiple URLs helps because not all of the generated URLs would be valid. Furthermore, by specifically generating *diverse* URLs, we also increase the coverage of content by the URLs. We empirically assess the impact of generating multiple URLs in Section 4.4.

We consider two variants to generate the $m$ URLs. LLM-CITE(DIVERSE) directly prompts the LLM to generate upto $m$ diverse URLs (prompt in Table 5). Each few-shot example in the prompt also includes upto $m$ URLs . LLM-CITE(CONTROLLED) prompts the LLM to generate only one URL for a given claim (prompt in Table 4), and then uses different decoding techniques to obtain $m$ URLs — beam search decoding (Sutskever et al., 2014), or sampling if beam search is unavailable for the model. Section 3.2 discusses these settings in more detail.

### 2.2 ON-THE-FLY FETCHING

Once we have a set of candidate citation URLs, the next step is to *fetch* the latest contents of the URL or lookup in a URL keyed content cache. One can use `curl`, `wget`, or equivalent tools for this to fetch in real time. This step allows LLM-CITE to have the most up-to-date information, and is thus key to verifying fresh claims. In this work, we focus on fact verification based on Wikipedia pages, and thus use the Wiki-API[2] to fetch contents of Wikipedia pages on-the-fly.

We emphasize that in contrast to fact verification systems that rely on an external search system, LLM-CITE relies only on a page retrieval system. Consequently, we avoid the expensive steps of indexing, embedding, query based retrieval and ranking, along with the associated engineering costs; see Section 4.5 for a detailed cost comparison. Furthermore, using URLs as the key for documents allows our method to take advantage of URL aliases and redirection. As mentioned in Section 2.1, some of the URLs generated by the LLM may be "guessed" based on the named entities in the claim. For instance, for the claim 'The scientific name for lady's fingers is Abelmoschus esculentus',

---

[2]https://www.mediawiki.org/wiki/API:Main_page

| | Does not require external index? | Can verify fresh claims? | Attribution? | Cheap? |
|---|---|---|---|---|
| FACTSCORE (Min et al., 2023) | ✗ | ✗ | ✓ | ✓ |
| LLMs + Web Search (Vu et al., 2024) | ✗ | ✓ | ✓ | ✗ |
| P(TRUE) (Kadavath et al., 2022) | ✓ | ✗ | ✗ | ✓ |
| LLM-CITE (ours) | ✓ | ✓ | ✓ | ✓ |

Table 1: Comparison of LLM-CITE with existing methods for fact verification. LLM-CITE combines the best of all desired traits and is simple, cheap, provides attribution and can verify fresh claims without requiring external search.

the LLM may generate the URL `https://en.wikipedia.org/wiki/Abelmoschus_esculentus` which redirects to the actual URL `https://en.wikipedia.org/wiki/Okra`.

## 2.3 NATURAL LANGUAGE INFERENCE

The final step in LLM-CITE is to check if the claim is entailed by any of the fetched documents. For this, we use an off-the-shelf natural language inference (NLI) model, which checks whether a hypothesis text is entailed by a premise text. A claim is considered verified if any of the documents entail it, and the URL(s) of the entailing document(s) is returned as the attribution. If none of the documents entail the claim, then the claim is not verified.

NLI is a well-studied NLP task with several public labeled datasets, and several off-the-shelf models; we refer the reader to Storks et al. (2019) for a survey. LLM-CITE can work with any NLI model. In our experiments, we use the Check Grounding NLI API[3] offered by Google Cloud, and also study the impact of using stronger LLM-based NLI models in Section 4.4.

NLI models typically work well on shorter premises (e.g. up to a few sentences long), whereas the premises in our case are a set of (potentially long) documents. To counter this mismatch, as a first step, we retrieve relevant sentences from each of the fetched documents. For this, we rely on a simple TF-IDF based retrieval (Cohen et al., 2003) to retrieve a total of $l$ sentences from every document. If $s_1, s_2, .., s_l$ represent the sentences in decreasing order of similarity to the claim, we create $l$ different premises for each document — $s_1$, $s_1|s_2$, $s_1|s_2|s_3$, $\cdots$, where | represents the concatenation operation.. We then query the NLI API model with all $l * m$ premises and the claim as the hypothesis.[4]

## 3 EXPERIMENTAL SETUP

We now explain our experimental setup for evaluating LLM-CITE, and comparing it against different fact verification methods. For all datasets, we focus on fact verification against Wikipedia pages, i.e., a claim is verified only if it is supported by any current page on Wikipedia.

## 3.1 DATASETS

We use three datasets to test three aspects of fact verification — long answers broken into atomic claims (Biographies), claims generated by diverse models (ASQA), and fresh claims (FreshQA).

**Biographies (Min et al., 2023).** This dataset contains long biographies (avg. 6 to 10 sentences) of people generated using LLMs. We use the subset of claims generated by InstructGPT (Ouyang et al., 2022). The dataset also provides human annotations of the long answer broken down into a list of atomic claims, along with a binary human judgement of whether the atomic claim is supported in Wikipedia — we use these binary judgements to evaluate different fact verification methods. We observed that some of the sentences, especially human atomic claims occurring in the later part of

---

[3]https://cloud.google.com/generative-ai-app-builder/docs/check-grounding

[4]Check Grounding API allows up to 200 facts (or premises) per hypothesis and only charges based on number of characters in the hypothesis.

the LLM's response, often don't have co-references resolved (e.g. 'He was a actor') which makes it hard to evaluate on those claims. In contrast, the first sentences of the biographies can be interpreted stand-alone. We thus use atomic claims corresponding to the first sentence in 100 biographies — the total dataset contains 443 claims which we use to benchmark different fact verification methods.

**ASQA (Stelmakh et al., 2022).** This dataset contains a subset of Natural Questions (Kwiatkowski et al., 2019) with human written answers along with the Wikipedia URLs used by the annotators to write the answer. This dataset has been used in prior work on generating citations (Gao et al., 2023). We use this dataset to test if fact verification methods can verify claims generated by diverse models. To do so, we use three models to generate answers for the questions in ASQA dataset — Gemma-1.1 7B instruction tuned model,[5] Gemini 1.0 Pro (Google, 2024) and PaLM 2 M (Anil et al., 2023). For all models, we instruct the LLM to generate complete sentences as the answers. We use greedy decoding to generate the answers. For the model generated answers, we don't have ground truth on whether they are factual. To circumvent this, we make the assumption that the human written answers are factual and attributable to the provided sources. We then filter the model generated answers and only keep those which are entailed by the human written answers (as determined by using the NLI model in 2.3). This ensures that the filtered answers are factual, and attributable to the same sources as the corresponding human written answers. In total, after filtering, we have 212 claims across the three models (86 Gemma-1.1 7B, 72 Gemini 1.0 Pro, 54 PaLM 2 M).

**FreshQA (Vu et al., 2024).** This dataset contains questions broken into categories depending on how frequently the answers change – never-changing, slow-changing, and fast-changing. In order to specifically check if fresh claims can be verified by LLM-CITE, we use questions from the fast-changing (77% in our data) and slow-changing subsets (23% in our data). Unfortunately, the answers to these questions provided in the dataset are sometimes outdated, and are often short answers. To resolve this issue, we manually write answers to 30 questions from this dataset based on current Wikipedia information. We then use these 30 answers to test whether various fact verification methods can verify them (Figure 1 (right) and Figure 3 (right) are two examples of fresh claims).

### 3.2 LLM-CITE SETUP

For our method, we experiment with using four different LLMs to generate the candidate citation URLs — Gemma 1.1 7B instruction tuned, Gemini 1.5 Flash (Reid et al., 2024), Llama2-7B-chat[6] (Touvron et al., 2023), and GPT-4o (OpenAI, 2023). We use $m = 4$ URLs for each claim. Unless otherwise specified, we use the variant LLM-CITE(DIVERSE) from Section 2.1 that prompts the LLM to generate 4 diverse URLs. We also experiment with the variant LLM-CITE(CONTROLLED), which prompts the LLM to generate one URL at a time, and uses beam search to obtain $m$ URLs per claim. Since Gemini 1.5 Flash and GPT-4o do not support beam search, we instead generate 4 URLs using sampling with temperature 0.8 for LLM-CITE(CONTROLLED). For the NLI model, we use $l = 6$ sentences and set the citation threshold to $0.6$. These parameters were tuned using the human written answers for the ASQA dataset.

**Rejection Sampling.** Since not all the generated URLs would be valid, we also explore rejection sampling to increase the percentage of valid URLs. Specifically, we use LLM-CITE(DIVERSE) variant with temperature 0.8 to sample a total of 5 responses. We only keep the responses which have the maximum number of valid URLs, and drop the other 4 sampled responses. We use the Wiki-API to determine the validity of the generated URLs. Note that using rejection sampling presents a trade-off between cost and URL validity. This method is denoted by LLM-CITE (RS).

### 3.3 BASELINES

**FACTSCORE (Min et al., 2023).** We use the state-of-the-art Gecko-1B embedding model (Lee et al., 2024) to build an index for a small subset of Wikipedia containing ≈8k documents — we ensure that relevant documents useful to verify the claims in any of the three datasets are included in this subset of Wikipedia. Note that using a small subset instead of the entire Wikipedia corpus

---

[5]https://huggingface.co/google/gemma-1.1-7b-it

[6]https://huggingface.co/meta-llama/Llama-2-7b-chat-hf

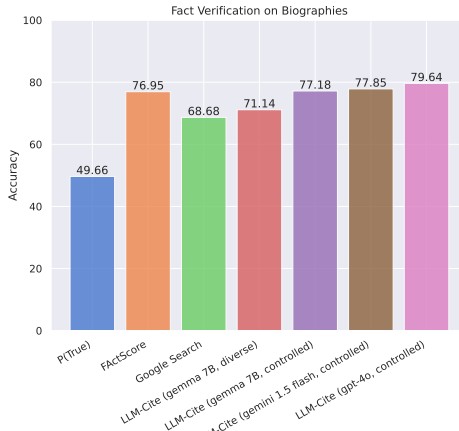 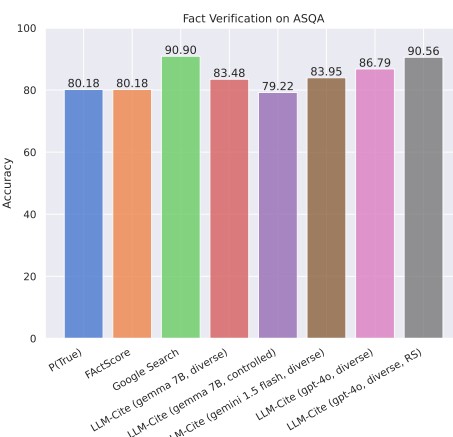

Figure 2: (left) Verification accuracy for human written claims obtained from long answers in Biographies dataset; (right) Verification accuracy with model generated claims in the ASQA dataset ; LLM-CITE performs comparable or better than existing methods across both datasets.

(≈6M documents) implies that the baseline results we report for FACTSCORE may overestimate the performance (since we use prior knowledge of which subset is important). For fair comparison to LLM-CITE, we use the same NLI model (as in Section 2.3) for FACTSCORE after the retrieval step.

**P(TRUE) (Kadavath et al., 2022).** This baseline relies on an LLM's parametric knowledge to judge the validity of a claim. Note that despite LLMs being pretrained without explicit labels for truthfulness, LLMs have been shown to be able to judge the truthfulness of a statement (Joshi et al., 2023). We design a 4-shot prompt where half of the examples are true claims and half are claims that are not true. The exact prompt can be found in Table 3. We parse the model generated response, and extract the model's judgement depending on if 'True' or 'False' is contained in the response. Note that this method does not provide any attribution as shown in Table 1. For all experiments, we use the Gemma 1.1 7B instruction tuned model and greedy decoding to generate the response, similar to the no-context baseline in Min et al. (2023).

**Google Search + NLI.** Since FACTSCORE uses a static index, by design it can't verify fresh claims. Hence, we also implement a baseline which first uses Google Search API[7] to retrieve top 4 results (restricted to Wikipedia only) with the claim as the query, and then uses the NLI model to check if the claim is entailed. We use the same NLI model from Section 2.3 for fair comparison. We want to note that even though google search can help retrieve fresh contents, it is very costly. See Section 4.5 for a more detailed discussion on the cost differences.

## 4 RESULTS

In this section, we compare LLM-CITE with existing fact verification methods across all the datasets. In all cases, for LLM-CITE we report results using at least one public model (e.g. Gemma 7B) and one private model (e.g. Gemini 1.5 Flash). Complete results are reported in Appendix A.3.

### 4.1 VERIFYING HUMAN-WRITTEN CLAIMS

Figure 2 (left) shows the accuracy of verifying human written atomic claims obtained from long answers using the Biographies dataset. Firstly, we observe that P(TRUE) does not perform well on this dataset, consistent with the findings from Min et al. (2023).[8] LLM-CITE performs competitively (and even marginally better) than the current best method FACTSCORE. We also observe that LLM-CITE with Gemma 7B performs very strongly, and is only marginally behind LLM-CITE with GPT-4o.

---

[7]https://developers.google.com/custom-search/v1/overview

[8]Note that we use few-shot prompting whereas their baseline uses 0-shot prompting.

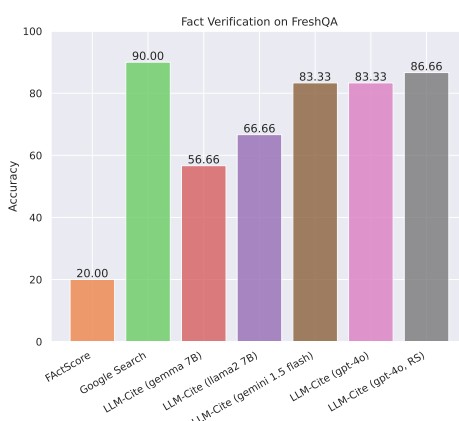
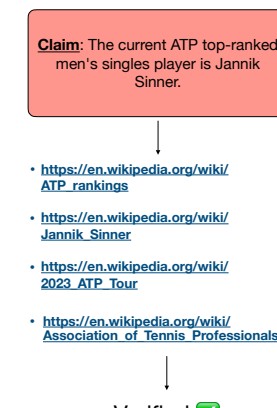

Figure 3: (left) Verification accuracy for manually written fresh claims to questions from FreshQA ; (right) Example of LLM-CITE verifying a fresh fact whose validity is *unknown* to the underlying model generating the URLs.

Surprisingly, we find that the Google Search baseline does not perform that well on this dataset. On investigating this, we find that Google Search performs poorly on some biographies pertaining to rare entities. Appendix A.4 shows example queries where Google Search fails, despite the existence of Wikipedia pages with the named entities.

**Takeaway.** LLM-CITE can verify human-written claims better than all existing fact verification methods including Google Search.

### 4.2 VERIFYING MODEL-GENERATED CLAIMS

Figure 2 (right) shows the verification accuracy on the ASQA dataset which contains claims generated by different models. We observe that LLM-CITE performs better than P(TRUE) and FACTSCORE. Interestingly, we observe that LLM-CITE(DIVERSE) which generates all 4 URLs directly performs better than LLM-CITE(CONTROLLED) where the model is prompted to only generate 1 URL at a time. We hypothesize this happens because the prompt explicitly encourages the model to generate diverse URLs, whereas the generated URLs are not as diverse in LLM-CITE(CONTROLLED).

We also observe significant gains when using rejection sampling with LLM-CITE instantiated with GPT-4o. With rejection sampling, the performance of LLM-CITE matches the performance of the Google Search baseline. Rejection sampling helps increase the percentage of valid URLs, thus both increasing the coverage of content, and being more robust to errors by the downstream NLI model.

**Takeaway.** LLM-CITE can verify claims generated by diverse models better than FACTSCORE and P(TRUE). It can match the performance of Google Search with rejection sampling.

### 4.3 VERIFYING FRESH CLAIMS

Figure 3 (left) compares LLM-CITE(DIVERSE) with baselines on the FreshQA dataset.[9] Unsurprisingly, we observe that FACTSCORE, which uses a static corpus (with cutoff before 2024), cannot verify fresh claims and only gets an accuracy of 20%. In contrast, LLM-CITE can verify fresh claims much better, with an improvement in performance as we move the underlying LLM (used for URL generation) from Gemma 7B to Llama2 7B to Gemini 1.5 Flash and GPT-4o. However, we find LLM-CITE to be behind the Google Search baseline, which performs the best on this dataset.

To understand the gap in performance, we manually analyze outputs and conclude that — (a) Google Search by design only retrieves valid URLs whereas 23% of the URLS generated by LLM-

---

[9]We do not evaluate P(TRUE) on this dataset since by design it cannot verify claims outside the LLM's knowledge cutoff better than random guessing.

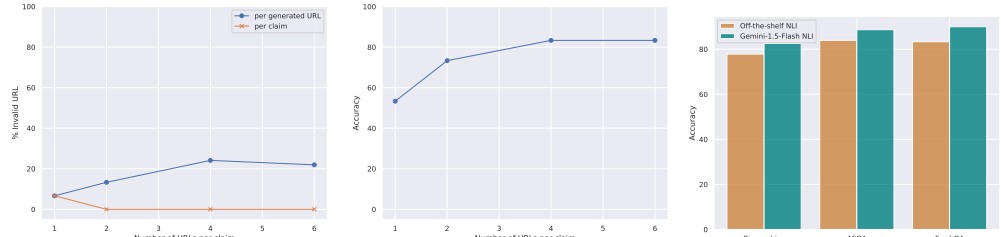

Figure 4: (left) % of invalid URLs with increasing number of generated URLs per claim — the % of examples where all generated URLs are invalid quickly drops to 0. ; (middle) Verification accuracy with increasing number of generated URLs. Performance increases significantly from 1 to 4 but plateaus afterwards. ; (right) Instead of using an off-the-shelf NLI model, using Gemini-1.5-Flash to do NLI improves performance across datasets ; All results are using LLM-CITE with Gemini-1.5-Flash on the FreshQA dataset.

CITE(DIVERSE) are invalid (see Section 4.4 for more results on URL validity). Thus, it gets more shots at recalling a corroborating document for fresh claims. The gap is further amplified by the fact that fresh claims typically have very few corroborating documents in the corpus. (b) FreshQA subset used here often relies on popular entities (e.g. Lionel Messi, Taylor Swift etc.) on which Google Search performs very well. This is in contrast to rarer entries present in the Biographies dataset (Figure 2 (left)) where we see that Google Search performs worse than LLM-CITE.

To address the errors in (a), we also explore using rejection sampling in LLM-CITE with GPT-4o since it reduces the percentage of invalid URLs. We observe that LLM-CITE with rejection sampling gets very close to the performance of Google Search. This is remarkable as the underlying LLM has not been exposed to information about the fresh claims in it's pretraining data.

**Takeaway.** LLM-CITE can verify fresh claims which is lacking in static methods like FACTSCORE. It gets close to performance of Google Search with stronger models and rejection sampling.

## 4.4 ANALYSIS OF LLM-CITE

**Increasing number of URLs** We study the benefit of generating more than one URL per claim in LLM-CITE. We do so by evaluating LLM-CITE(DIVERSE) with $m$ (no. of generated URLs) in $[1, 2, 4, 6]$. In all experiments, we use Gemini-1.5-Flash to generate URLs. We modify the prompt for each $m$ so that the instructions ask for $m$ URLs and each few-shot example includes up to $m$ URLs.

*Generating more URLs significantly decreases the chance that none of the URLs are valid.* Figure 4 (left) shows — (a) percentage of invalid URLs among all generated URLs across all claims (orange) ; (b) percentage of claims where all the generated URLs for that claim are invalid (blue). We observe that as the number of generated URLs increases, the number of examples where none of the URLs are valid quickly drops to zero. This implies that we always have documents available to check the validity of the claim in the final step of LLM-CITE. Furthermore, even when generating up to 6 URLs, less than 25% of URLs are invalid. This allows LLM-CITE to have access to diverse, valid URLs in further steps of the method.

*Generating more URLs increases downstream verification performance.* Figure 4 (right) shows the fact verification accuracy with increasing number of URLs. All results here use LLM-CITE with Gemini-1.5-Flash to generate URLs. We observe that performance increases significantly as the number of generated URLs increases from 1 to 4, but plateaus afterwards. This plateau is expected, since for short atomic claims there might not be enough diverse Wikipedia URLs available to verify the claim (we also noticed this when designing the few-shot prompts).

**LLM instructed for NLI.** As discussed in Section 2.3, LLM-CITE uses an off-the-shelf NLI model. Since the NLI model works better on shorter premises, we had to use a sentence retriever to select relevant sentences from fetched documents to form the (short) premise. Through manual error analysis, we observed instances where the generated URL in LLM-CITE was correct, but the NLI model (or sentence retrieval) had errors.

| Method | API Cost / 1000 queries |
|---|---|
| Google Search + NLI[11] | $5 |
| LLM-CITE(DIVERSE) | $0.0513 |
| LLM-CITE(CONTROLLED) | $0.0513 |
| LLM-CITE(DIVERSE) + Rejection Sampling | $0.1509 |

Table 2: Cost comparison of Google Search vs URL generation in LLM-CITE. All rows for LLM-CITE use Gemini-1.5-Flash for generating URLs and are computed based on 352 input tokens and 83 outputs which is the average for FreshQA.

To address this shortcoming, we instructed an LLM, Gemini-1.5-Flash, to perform the NLI task. Specifically, given a document and claim, we instruct the model to verify if the claim is true or false relative to the given document. Table 6 shows the prompt. Note that this method does not require any prior sentence retrieval, and is similar to the validation step in Min et al. (2023). Figure 4 (right) shows a comparison across all the three datasets, where we use Gemini-1.5-Flash to generate URLs in our method. We observe that using an LLM instructed for NLI consistently improves performance over an off-the-shelf NLI model. We would like to note this increase in performance comes at increased cost, as the LLM-based NLI variant is significantly more costly than the Check Grounding API ($0.32 vs $0.05 per 1000 claims).[10] See Appendix A.2 for more details.

## 4.5 COST COMPARISON

We compare the cost of LLM-CITE with other methods to do fact verification such as Google Search + NLI. The cost of LLM-CITE consists of two main parts — URL generation, and entailment checks using the NLI model. Since the NLI part is common across LLM-CITE and Google Search + NLI baseline, we compare the cost of URL generation in LLM-CITE with Google Search. Methods like Google Search that rely on a search index have a lot of upfront engineering cost associated with setting up an index, retrieval and ranking models. These costs are not present for LLM-CITE.

We further compare the inference times costs of LLM-CITE and Google Search. Direct comparison of the computational cost (say, using FLOPs, which are commonly used for measuring LLM computation costs (Kaplan et al., 2020)) is difficult since the underlying infrastructure and algorithms are very different. Consequently, we compare the two methods via API costs, which typically account for the engineering and maintenance costs as well.

To compute cost for URL generation in LLM-CITE, we compute the average number of input and output tokens for each model separately using the FreshQA dataset and use the published per token costs to compute the total cost. Table 2 reports this cost for different variants of LLM-CITE with Gemini 1.5 Flash, along with the published API cost for Google Search.[12]

URL generation with Gemini 1.5 Flash in LLM-CITE(DIVERSE) is more than 90x cheaper than Google Search, and more than 30x cheaper with rejection sampling. Incorporating NLI costs (Appendix A.2), LLM-CITE(DIVERSE) is overall more than 45x cheaper than Google Search + NLI. The cost of LLM-CITE may differ if we use a different model for URL generation. For instance, for Gemma 7B, using standard costs for a 7B model[13], the cost for URL generation per 1000 queries is $0.089 (353 input tokens and 93 output tokens). For GPT-4o, the cost is $2.5 per 1000 queries (293 input and 68 output tokens).[14]

## 5 RELATED WORK

**Fact Verification.** Datasets such as PolitiFact (Vlachos & Riedel, 2014), RumourEval (Derczynski et al., 2017), FEVER (Thorne et al., 2018), and SciFact (Wadden et al., 2020) have been used for fact

---

[10]https://cloud.google.com/generative-ai-app-builder/pricing#check_grounding_api_pricing
[11]https://developers.google.com/custom-search/v1/overview
[12]https://cloud.google.com/vertex-ai/generative-ai/pricing
[13]https://www.together.ai/pricing
[14]https://openai.com/api/pricing/

verification where each claim has to verified against a corpus like Wikipedia or a scientific corpus. Hanselowski et al. (2019) provides a detailed overview of existing fact verification datasets. The main difference is that they focus on human written claims only, whereas our work (and more recent prior work) also focus on evaluating the factuality of LLMs' responses.

A lot of existing methods for evaluating factuality rely on retrieval — either web search (Popat et al., 2017; Wei et al., 2024) or retrieval with a static corpus (Lewis et al., 2020; Min et al., 2023). Another line of work relies on LLM's parametric knowledge for fact verification — either simple prompting (Lee et al., 2020), or few-shot prompting (Lee et al., 2021; Kadavath et al., 2022), or hierarchical prompting (Zhang & Gao, 2023), or self-checking (Manakul et al., 2023).

**Generative Retrieval.** The main idea behind LLM-CITE is that we can use the LLM itself to generate URLs i.e., an index for documents. Similar ideas have been explored especially for retrieval in retrieval-augmented generation (RAG) — Tay et al. (2022) finetune transformers to directly generate index for retrieval, and Wang et al. (2022) also propose a neural network to directly generate document identifiers. The main difference is that these methods require *finetuning* a model for the task, whereas we do not require any finetuning and show that we can leverage the *pretrained knowledge* to directly generate URLs. Additionally, these works propose a variant of RAG where the retrieval is done using a model, whereas our goal is to do fact verification with attribution. Recently, Khalifa et al. (2024) propose to pretrain models with source identifiers which can enable generation with attribution — they focus on a synthetic setting and require both pretraining and instruction tuning. In contrast, we explore diverse real world claims and do not require any further training to generate attribution.

# 6 DISCUSSION

We highlight few key aspects about LLM-CITE. First, the method is general and is not constrained to use Wikipedia only. It can be easily extended to any other reliable domains appearing on the Web (e.g., `nytimes.com`) that are also seen during pretraining. Second, identifying documents by their URLs is crucial to the success of LLM-CITE. URLs benefit from both being seen during pretraining and also having a semantic structure that enables generalization. Given this, we expect stronger, larger pretrained models to perform better as they would memorize a large number of URLs seen during pretraining and their superior language understanding would help them form stronger associations between claims and URLs. Lastly, even though we use on-the-fly fetching using the URLs, in practice one would use a cached key-value lookup store to reduce end-to-end latency and host load. LLM-CITE can be implemented with under 1s latency per claim (1 ms per token for URL generation and under 500ms for NLI) and we expect this to improve further as hardware and models get better.

**Future Work.** Here we describe three key future directions for our work. (1) In this work, we leverage the pretraining knowledge and directly prompt LLMs to generate the URLs. However, we expect that performance can be further improved by instruction tuning LLMs on the 'URL elicitation' task. (2) The current proposed method is best suited for fact verification against public domains seen in the pretraining data. To extend the approach to private corpora, one would need to perform additional finetuning to teach the model the association between a document identifier and its contents. This is similar to methods such as source-aware training (Khalifa et al., 2024). (3) Even for domains seen during pretraining, not all would have meaningful URLs (e.g. arxiv.org only has number identifiers). In such cases, instead of using URLs we can create a cached key-value lookup with more meaningful identifiers e.g. title. We leave this exploration for future work.

# 7 CONCLUSION

We present a very simple, cheap, and easy-to-implement method for fact verification, LLM-CITE, that does not require any external retrieval or web search. LLM-CITE off-loads the expensive task of searching through a corpus to the LLM, by directly generating candidate citation URLs. LLM-CITE performs on par or better than all existing methods for fact verification, across different types of claims (fresh, non-fresh, model generated, human written) — while still providing attribution and being very cheap.

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

# A APPENDIX

## A.1 PROMPTS

Here, we describe all the prompts used in this work. For Gemini-1.5-Flash, we observed that the model response always began with preamble (e.g. 'Here are some URLs..') and included some explanation. In order to parse the outputs easily, we include an additional sentence in the instruction for Gemini-1.5-Flash — 'Directly generate the URL without any explanation'.

Table 3 is the prompt used for P(TRUE). Table 5 is the prompt used for our method LLM-CITE(DIVERSE). Table 4 is the prompt used for the variant of our method LLM-CITE(CONTROLLED). For the results in Section 4.4 where we directly use an LLM instructed for NLI intead of an off-the-shelf NLI model, we use the prompt in Table 6.

## A.2 COST COMPARISON: NLI

In this section, we describe the details of comparing the cost of using Check Grounding API (off-the-shelf NLI model) vs using an LLM instructed for NLI, as described in Section 4.4.

For Check Grounding API, note that we have a total of $m * l$ no. of premises — for each of the $m = 4$ documents we have $l = 6$ retrieved sentences used to create $l$ premises as described in Section 2.3. But the API allows up to 200 facts in a single call, and only charges based on characters in the claim/answer — \$0.00075 per 1k characters[15]. Hence we need only one API call per claim. Assuming 70 characters per claim (which is the average for FreshQA), the total cost for 1000 claims is $0.00075 * 70 = \$0.0525$.

---

[15] `https://cloud.google.com/generative-ai-app-builder/pricing#check_grounding_api_pricing`

---

Instruction: Given a claim your task is to verify if it is true or false. For example,

Claim: New York City is the capital of the USA.
Answer: False

Claim: Virat Kohli is an Indian cricketer.
Answer: True

Claim: The 2000 summer olympics were held in Miami.
Answer: False

Claim: Russia is the largest country in the world.
Answer: True

Claim: {claim}
Answer:

---

Table 3: 4-shot prompt used for P(TRUE).

---

Instruction: Given a sentence, generate a wikipedia URL that can help verify the sentence. For example:

Sentence: Virat Kohli is an Indian cricketer.
URL: https://en.wikipedia.org/wiki/Virat%20Kohli

Sentence: The population of Washington DC is more than 600,000.
URL: https://en.wikipedia.org/wiki/Washington,%20D.C.

Sentence: On February 2015, the Super Bowl XLIX was played in Glendale, Arizona.
URL: https://en.wikipedia.org/wiki/Super%20Bowl%20XLIX

Sentence: {claim}
URL:

---

Table 4: Instruction used for LLM-CITE(CONTROLLED) to generate URLs for a given claim. Note that for Gemini-1.5-Flash, we include an additional sentence in the instruction: 'Directly generate the URL without any explanation'.

For Gemini-1.5-Flash instructed for NLI, we set the maximum number of characters in each document to be 4000 (roughly 1000 tokens). Note that the number of output tokens is significantly smaller (only 'True' and 'False') than number of input tokens. Including the characters in instruction and the claim, the total no. of input tokens is 4257. Considering a cost of \$0.00001875 per 1k input characters[16], and $m = 4$, the total cost is $0.00001875 * 4.257 * 1000 * 4 = $ \$0.3192 per 1000 claims.

## A.3 LLM-CITE RESULTS ACROSS MODELS

In Section 4, we report the results of using LLM-CITE with a smaller model (e.g. Gemma 7B) and a larger model (e.g. GPT-4o). Here, we report the complete results of using all four models for URL generation (Gemma 1.1 7B instruction tuned, Llama2-7B, Gemini 1.5 Flash, GPT-4o) on all three datasets (Biograohies, ASQA, and FreshQA). Table 7 shows the results.

## A.4 ERROR ANALYSIS: GOOGLE SEARCH

The results in Section 4.1 for the biographies dataset showed that Google Search does not perform well on this dataset, in contrast to the other datasets we study in this paper. To investigate why

---

[16]https://cloud.google.com/vertex-ai/generative-ai/pricing

Instruction: Given a sentence, generate upto 4 diverse wikipedia URLs that may help to verify the given sentence. For example:

Sentence: Virat Kohli is an Indian cricketer.
URLs: https://en.wikipedia.org/wiki/Virat%20Kohli
https://en.wikipedia.org/wiki/India%20national%20cricket%20team
https://en.wikipedia.org/wiki/Career%20of%20Virat%20Kohli

Sentence: On February 2015, the Super Bowl XLIX was played in Glendale, Arizona.
URLs: https://en.wikipedia.org/wiki/Super%20Bowl%20XLIX
https://en.wikipedia.org/wiki/2014%20New%20England%20Patriots%20season
https://en.wikipedia.org/wiki/2014%20Seattle%20Seahawks%20season

Sentence: The population of Washington DC is more than 600,000.
URLs: https://en.wikipedia.org/wiki/Washington,%20D.C.
https://en.wikipedia.org/wiki/Demographics%20of%20Washington,%20D.C.
https://en.wikipedia.org/wiki/Washington%20metropolitan%20area
https://en.wikipedia.org/wiki/Timeline%20of%20Washington,%20D.C.

Sentence: {claim}
URLs:

Table 5: Instruction used for LLM-CITE(DIVERSE) to generate URLs for a given claim. Note that for Gemini-1.5-Flash, we include an additional sentence in the instruction: 'Directly generate the URL without any explanation'.

Instruction: Given a document and a claim, your task is to check if the claim is true or false given that document. Directly generate True or False without any explanation.

Document: {doc}

{claim} True or False?

Table 6: Prompt used to perform NLI with Gemini-1.5-Flash.

this happens, we manually analyzed outputs from the Google Search results and found poor results potentially due to the rare nature of entities. We include some examples here for illustration:

1. site:wikipedia.org Hesham Nazih is an entrepreneur.
   – The top returned URLs include: `https://en.wikipedia.org/wiki/Cinema_of_Egypt` and `https://en.wikipedia.org/wiki/List_of_Syrians` but does not include the correct URL useful for `https://en.wikipedia.org/wiki/Hesham_Nazih`.

2. site:wikipedia.org Lily Branscombe is an entrepreneur.
   – The top returned URLs include `https://en.wikipedia.org/wiki/Women_in_aviation`, `https://en.wikipedia.org/wiki/Wikipedia:WikiProject_Biography/Article_alerts/Miscellaneous/Archive_5` and `https://de.wikipedia.org/wiki/Wikipedia:WikiProjekt_Frauen/Frauen_in_Rot/Fehlende_Artikel_nach_NationalitÃd't/NeuseelÃd'nderinnen` (which does mention the person), but does not include the correct URL useful for verification `https://en.wikipedia.org/wiki/Lily_Branscombe`.

3. site:wikipedia.org Regina Martínez Pérez specialized in reporting on politics.
   – The top returned URLs include `https://en.wikipedia.org/wiki/Kidnapping_and_murder_of_MoisÃl's_SÃąnchez_Cerezo`, `https://en.wikipedia.org/wiki/Maria_Moors_Cabot_Prizes`, and `https://en.wikipedia.org/wiki/List_of_unsolved_murders_(2000âĂŞpresent)` which do mention the person sometimes but are not the main page for the person `https://en.wikipedia.org/wiki/Regina_MartÃ■nez_PÃl'rez`.

| Model | Biographies | ASQA | FreshQA |
|---|---|---|---|
| Gemma 1.1 7B it | 77.18 | 83.48 | 56.66 |
| Llama2-7b-chat | 77.40 | 83.48 | 66.66 |
| Gemini 1.5 Flash | 77.85 | 83.95 | 83.33 |
| GPT-4o | 79.64 | 86.7 | 83.33 |

Table 7: Fact verification across all models and all datasets for LLM-CITE. For ASQA and FreshQA, we use LLM-CITE(DIVERSE) whereas for Biographies we report LLM-CITE(CONTROLLED) since that performed better.