# OpenReview forum: "LLM-Cite: Cheap Fact Verification with Attribution via URL Generation"
_ICLR.cc/2025/Conference — Submitted to ICLR 2025_

### Official Review · Reviewer_493p · 2024-10-22

**Soundness:** 2
**Presentation:** 3
**Contribution:** 2
**Rating:** 3
**Confidence:** 4

**Summary:**

The paper introduces LLM-Cite, a novel method to address hallucinations in Large Language Models (LLMs) by verifying the factuality of their responses without reliance on costly external search systems or static knowledge bases. Instead of depending on frequent updates of knowledge bases or LLM's internal parametric knowledge—which lacks attribution—LLM-Cite leverages the LLM itself to generate potential citation URLs for claims. These URLs are fetched on-the-fly, and the claims are verified using entailment checks against the URL content. This approach ensures both attribution and the ability to verify current information, significantly reducing costs compared to traditional search methods. The authors evaluate LLM-Cite on three datasets encompassing both fresh and non-fresh claims from human and model-generated sources, demonstrating its competitive or superior performance compared to existing methods while being more than 45 times cheaper than a Google Search-based approach.

**Strengths:**

- The proposed method is neat, simple, and interesting.
- The paper is easy-to-read overall.

**Weaknesses:**

- Restricted to Wikipedia: Although the authors demonstrate the proposed fact verification approach is effective, the experiments are only conducted on Wikipedia. There are concerns whether this method could be extended to beyond this limited knowledge base.
- Dataset issue: First, while the author tested on three datasets, most of these dataset are relatively small. In particular, the FreshQA dataset only has 30 instances, which raises concerns about generalizability and whether the conclusion still holds if we increase the size of the testing data. Second, for the ASQA dataset, the author uses a NLI model to ensure data quality. However, as seen in Figure 4 right, the NLI model performance is only somewhere around 85%. The chances that the NLI model makes an error is higher than the performance gap between the proposed approach and the baseline methods (Figure 2 right). Hence, the conclusion from Figure 2 right may be questionable. Third, for the Biographies dataset, the author has transformed the data in a way that is more suitable for their own approach, which may lead to unfair comparisons with baselines (see below).
- Unfair baseline comparison: First of all, the authors did not use the same LLM for their method and baseline method, which is unfair. Additionally, for FactScore, the whole point of the paper is decomposing long texts into atomic facts and then evaluating factuality. However, due to the experimental setup, this decomposition part seems removed. Therefore, it may be unfair for the authors to say they “beat the current best method (FactScore) on Biographies”.

**Questions:**

- NLI choice: For the NLI models, there are existing models that are designed to tackle the mismatch in granularity between premise and hypothesis, such as DocNLI and AlignScore. These are naturally more suited for the task. I was wondering why these were not considered?
- Citation threshold: What does this mean? Does it mean > 0.6 is entailment?

---

> ### Author Response · Authors · 2024-11-21
> **Response to Reviewer 493p**
>
> We thank the reviewer for the insightful comments and feedback!
>
> >  Restricted to Wikipedia: Although the authors demonstrate the proposed fact verification approach is effective, the experiments are only conducted on Wikipedia. There are concerns whether this method could be extended to beyond this limited knowledge base.
>
> We have tried to address this in the general response.
>
> > Unfair baseline comparison: First of all, the authors did not use the same LLM for their method and baseline method, which is unfair. Additionally, for FactScore, the whole point of the paper is decomposing long texts into atomic facts and then evaluating factuality. However, due to the experimental setup, this decomposition part seems removed. Therefore, it may be unfair for the authors to say they “beat the current best method (FactScore) on Biographies”.
>
> All the baselines and our method use the same NLI model. For P(True), we used gemma 7B and all result figures also include a comparison to our method with gemma 7B for URL generation. Compared with the same model, LLM-Cite is better than P(True) across all datasets while also providing attribution which is lacking in P(True).
>
> We do agree that breaking down long texts into atomic facts is an important point in FactScore. What we are showing is that after breaking down into atomic facts, we can do URL-generation instead of retrieval and it performs equally well if not better, while having other benefits like cheaper and freshness. So the comparison is to the retrieval part in FactScore while keeping the other components the same (breaking down into atomic claims, and NLI). In fact, the error rate reported in table 3 in the FactScore paper is over atomic claims — exactly the same metric we report (1 - error rate) in our results.
>
> > Dataset issue: First, while the author tested on three datasets, most of these dataset are relatively small.
>
> We do agree that our datasets are not very big — for FreshQA specifically, we used a smaller set since we manually wrote answers to make sure they are up to date.
>
> > Second, for the ASQA dataset, the author uses a NLI model to ensure data quality. However, as seen in Figure 4 right, the NLI model performance is only somewhere around 85%.
>
> We want to clarify that the 85% number shown in Figure 4 is the end-to-end verification accuracy and not NLI accuracy — the 15% error includes cases where the URL generated was valid but not useful to verify the claim, and hence is not always NLI error.
>
>
> > Citation threshold: What does this mean? Does it mean > 0.6 is entailment?
>
> Citation threshold is used by Check Grounding, the NLI model we use for verification – https://cloud.google.com/generative-ai-app-builder/docs/check-grounding. It corresponds to a threshold for the confidence above which we say that the claim is entailed.

---

> > ### Comment · Reviewer_493p · 2024-11-25
> > **Response by Reviewer**
> >
> > Thank you for the response. The author response has addressed my concerns on unfair baseline comparison and citation threshold. However, issues remain for the size and quality of the dataset as well as the limitations with Wikipedia as the only evidence source. Therefore, I will keep my original ratings.

---

### Official Review · Reviewer_MBUm · 2024-11-05

**Soundness:** 2
**Presentation:** 3
**Contribution:** 2
**Rating:** 3
**Confidence:** 4

**Summary:**

This paper proposes a new method called LLM-CITE for factual verification tasks, aiming to leverage LLM's understanding of URLs during the pre-training phase to perform text factuality checking by utilizing generated URLs to retrieve evidence. Compared to existing methods, the proposed method is simple, cost-effective, and easy to deploy, without the need for complex external retrieval processes. The effectiveness of the proposed method has been validated on three datasets.

**Strengths:**

1. New insights: This paper suggests using LLM's understanding of URLs during the pre-training phase for evidence querying. The underlying assumptions are: 1. The structure of evidence URLs usually follows a pattern; 2. LLMs have been exposed to a large number of URL training instances during pre-training, thus they can generate new URLs for fresh claims.
2. This paper is well-written and easy to follow.

3. The model's effectiveness has been validated on human-written, model-generated, and fresh claims.

**Weaknesses:**

1. The paper's assumptions about URLs have limited generalizability. Although the paper clarifies that it can be applied to any web domain, intuitively, its application is primarily focused on verification that has a high degree of regularity in URL structure, such as Wikipedia. It is difficult to implement for real-world claim verification like PolitiFact or gossip, and it remains in an ideal stage.

2. Factual verification claims often exhibit high semantic complexity and typically require multi-hop evidence for verification, which the proposed method clearly struggles to achieve.

3. URLs are extremely sensitive to characters, and the paper does not handle invalid URLs reasonably and does not provide experimental results for valid URLs.

4. As a benchmark dataset for factual verification, why not validate on datasets like FEVER, HOVer, and FEVEROUS, which use Wikipedia as an external knowledge base?

**Questions:**

Weaknesses above.

---

> ### Author Response · Authors · 2024-11-21
> **Response to Reviewer MBUm**
>
> We thank the reviewer for the insightful comments and feedback!
>
> > The paper's assumptions about URLs have limited generalizability. Although the paper clarifies that it can be applied to any web domain, intuitively, its application is primarily focused on verification that has a high degree of regularity in URL structure, such as Wikipedia. It is difficult to implement for real-world claim verification like PolitiFact or gossip, and it remains in an ideal stage.
>
> We have tried to address the use of Wikipedia in the general response.
>
> > Factual verification claims often exhibit high semantic complexity and typically require multi-hop evidence for verification, which the proposed method clearly struggles to achieve.
>
> This is a valid point and we will acknowledge it in the paper. However, it is also an orthogonal issue in the following sense. Our key point is that LLM-cite + lookup could be a drop-in replacement for search+retrieval for fact verification. Both approaches would struggle equally with verifying claims requiring multi-hop reasoning. Furthermore, the datasets we use in this work are realistic datasets which have been used in prior work for fact verification. For larger claims (e.g. long answers in Biographies), each long response is first broken down into atomic claims which are less likely to require multihop reasoning.
>
> > URLs are extremely sensitive to characters, and the paper does not handle invalid URLs reasonably and does not provide experimental results for valid URLs.
>
> Section 4.4 gives a detailed analysis of when and how many of the generated URLs are invalid. We generate multiple URLs per claim, and there are close to 0% of claims where all of the generated URLs for the claim are invalid (Figure 4 left) — this means we always have some valid URL against which we verify the claim.
>
> > As a benchmark dataset for factual verification, why not validate on datasets like FEVER, HOVer, and FEVEROUS, which use Wikipedia as an external knowledge base?
>
> Our motivation was to show that for existing factuality evaluation methods like FactScore, we can replace the retrieval part with URL-generation and it does equally well (if not better) — hence we used the same dataset as in FactScore. To show that URL-generation gives additional benefits such as verifying fresh claims, we decided to use the FreshQA dataset.

---

### Official Review · Reviewer_6T35 · 2024-11-07

**Soundness:** 2
**Presentation:** 3
**Contribution:** 2
**Rating:** 3
**Confidence:** 4

**Summary:**

The paper presents LLM-Cite - an LLM-based fact verifier that bypasses the need for expensive retrieval-based fact-checking to guarantee freshness and, at the same time, does not rely on the un-updated "frozen" learning by the model. It introduces a URL-generation step where the sources of the premises are fetched, such that those premises can be used further to entail the fact/claim as a hypothesis using an off-the-shelf NLI model.

**Strengths:**

I have found the paper to be easy to read. The methodology is lucid and has been explained well. The experiment designs are adequate to establish the claim sufficiently. The findings are worth noting, and the overall framework can be of some value to developers during the selection of fact verification models. The work is original (regarding my limited scholarship in fact verification), given the use of the existing SOTA NLI model for fact (hypothesis) corroboration coupled with SOTA LLMs for URL generation to provide sources of the premises. Also, making the pipeline much cheaper (cost-wise) by avoiding retrieval-based NLI has a significant impact on the community.

**Weaknesses:**

Despite the idea being interesting, I have found some technical issues that weakened the overall soundness. I enumerate them as follows:

1. The assumption that generated URLs are always meaningfully related to the core content of the document from where the premises are to be fetched is not true by and large. It works for Wikipedia because the URLs are well-structured semantically.

2. LLMs generating URLs on Wikidata have a significantly higher probability of being linked with a valid URL because extensive entity linking has already been done. This, however, is not the case for many other web sources.

3. There are several URLs that are not named according to the premise entities. In that case, those sources will never be fetched.

4. How to resolve contradictory entailment from premises belonging to different sources?

5. There can be many sources that are themselves false (particularly for the open Internet and also in cases of unverified Wiki pages). So assuming the premises to be true may lead to incorrect RTE.

6.  It is unclear how the prompt templates are designed, i.e., the rationale and methodology that would drive the demonstration example patterns in the few-shot cases.

7. A discussion on the prompt dataset (for the few-shot case) creation together with its source should be discussed.

8. The assumption that RTE (i.e. NLI) being true would imply that the hypothesis (fact/claim) is verified is bit tricky and may not always be true. A false statement can entail a hypothesis as well as its true version. Eg.:

$\textit{Apples come in many colors}$  $\implies$ $\textit{Apples can be blue (claim)}$.

$\textit{John was a follower of Jesus}$  $\implies$ $\textit{John was a Christian (claim)}$.

9. Line 253: What is citation threshold? I could not find the definition.

10. In the comparisons with the baselines and variants of LLM-Cite, what was the justification behind not keeping the model set fixed for all the experiments? I think this should be clear.

11. In sections 4.1 and 4.2, an analysis of why the verification models perform better on model-generated claims as compared to human-generated claims is very important to me. I could not find any adequate analysis for that.

12. The key success of LLM-Cite depends on the NLI model (given that at least one valid URL is generated that points to a valid premise). Hence, a discussion on the accuracy of SOTA NLI models (with citation) and the rationale behind choosing the Check Grounding NLI API and Gemini-1.5-Flash should be included.

**Questions:**

Q-1: Lines 414-416 are not clear. Figure 4 (left) does not properly depict the description therein. In my understanding, the blue line depicts the % of URLs that are invalid among the generated k URLs, while the orange depicts the % of claims where all the URLs are invalid.

---

> ### Author Response · Authors · 2024-11-21
> **Response to Reviewer 6T35**
>
> We thank the reviewer for the insightful comments and feedback! One general point we wanted to bring up, especially in regards to the comments about NLI is that we build off of existing well-established methods such as FactScore, and our main idea is to replace the retrieval with URL generation (which has advantages as we demonstrated)  — we keep all other parts of the pipeline such as use of Wikipedia and NLI the same.
>
> We try to address some of the specific concerns which were raised:
>
> > The assumption that generated URLs are always meaningfully related to the core content of the document from where the premises are to be fetched is not true by and large. It works for Wikipedia because the URLs are well-structured semantically.
>
> > LLMs generating URLs on Wikidata have a significantly higher probability of being linked with a valid URL because extensive entity linking has already been done. This, however, is not the case for many other web sources.
>
> We have tried to address the concern regarding Wikipedia only in the general response.
>
> > There are several URLs that are not named according to the premise entities.  In that case, those sources will never be fetched.
>
> We have also tried to address this in the general response.
>
> > How to resolve contradictory entailment from premises belonging to different sources?
>
> > There can be many sources that are themselves false (particularly for the open Internet and also in cases of unverified Wiki pages). So assuming the premises to be true may lead to incorrect RTE.
>
> That is a really good point, and we followed other works such as FactScore in making the assumption about contradictory premises (assumption 3 in their paper section 3.1). In our method, even if any source entails the claim, it will be verified and because LLM-Cite gives attribution, the user can determine if it’s a trustworthy source (e.g. maybe a user is less likely to trust a wikipedia page which was not thoroughly verified). In general, we envision a setting where the user gives a set of documents that they trust, and our method can be constrained to use those documents only.
>
> > A discussion on the prompt dataset (for the few-shot case) creation together with its source should be discussed.
>
> Thanks for pointing this out, we will add more discussion on how we designed the few-shot prompt. We want to note that we did not actually do much prompt engineering — a simple, intuitive prompt with a few diverse examples already works well.
>
> > Line 253: What is citation threshold? I could not find the definition.
>
> Citation threshold is used by Check Grounding, the NLI model we use for verification – https://cloud.google.com/generative-ai-app-builder/docs/check-grounding. It corresponds to a threshold for the confidence above which we say that the claim is entailed.
>
> > In the comparisons with the baselines and variants of LLM-Cite, what was the justification behind not keeping the model set fixed for all the experiments? I think this should be clear.
>
> All the baselines and our method use the same NLI model. For P(True), we used gemma 7B and all result figures also include a comparison to our method with gemma 7B for URL generation. Compared with the same model, LLM-Cite is better than P(True) across all datasets while also providing attribution which is lacking in P(True).
>
> > In sections 4.1 and 4.2, an analysis of why the verification models perform better on model-generated claims as compared to human-generated claims is very important to me.
>
> This observation comes from our difference performance on ASQA (model generated claims) and Biographies (human generated claims). However, we did not want to call this strongly as the two datasetsalso differ in other things such as domain, average length of query etc, so there could be potential confounders that affect our performance on the claims.
>
> > The key success of LLM-Cite depends on the NLI model (given that at least one valid URL is generated that points to a valid premise). Hence, a discussion on the accuracy of SOTA NLI models (with citation) and the rationale behind choosing the Check Grounding NLI API and Gemini-1.5-Flash should be included.
>
> We will make sure to add more discussion around this in the paper. Check Grounding NLI model is a cheaper alternative to prompting larger models for entailment since it was specifically finetuned for NLI. However, we are consistent in using the same NLI model for all approaches (our method, and baselines) and don't expect trends to change with any other NLI model. We discuss this in section 4.4 where a stronger NLI model improves performance but is costlier than Check Grounding NLI model.

---

> > ### Comment · Reviewer_6T35 · 2024-11-25
> >
> > Thank you for your response. I have carefully gone through all the arguments. However, I would like to retain my position for the following reasons:
> >
> > 1. The work is too Wikipedia and FactScore centric, thereby leaving the question of why even to generate URL? Why not just retrieve entity information (premise statements) directly from Wikidata/DBPedia?
> >
> > 2. The possibility of confounders, as stated by the authors, make the results even more weak. I am wondering what could be the nature of some of these confounders.

---

> > > ### Author Response · Authors · 2024-11-26
> > >
> > > Thanks for going through our response!
> > >
> > > 1. As we mention in the general response, the URLs required to verify the claims go beyond just identifying the entity name e.g. the URL https://en.wikipedia.org/wiki/Demographics_of_the_world (which LLM-Cite can generate) is useful to verify the claim ‘The current population of the world is more than 8 billion as of 2024’ where the path name in the URL is not explicitly mentioned.  In other cases, different structures such as lists (e.g. https://en.wikipedia.org/wiki/List%20of%20New%20Zealand%20national%20cricket%20captains), history of entities (e.g. https://en.wikipedia.org/wiki/History%20of%20Krispy%20Kreme), etc. is also required to identify the correct URL to verify the claim.
> > >
> > > 2. We want to clarify the we did not want to make the following claim which was pointed out in the review, since it is comparing results on two different datasets.
> > >
> > > > an analysis of why the verification models perform better on model-generated claims as compared to human-generated claims is very important to me.
> > >
> > > Our results comparing LLM-Cite to other baselines are not affected by confounders --- all methods are evaluated on the same claims within each dataset, and we make sure to control for other things like NLI model which is same across all methods.

---

### Official Review · Reviewer_CMf6 · 2024-11-09

**Soundness:** 2
**Presentation:** 2
**Contribution:** 2
**Rating:** 3
**Confidence:** 5

**Summary:**

The paper proposes LLM-Cite, a cheap and easy approach to leverage an LLM to directly generate potential citation URLs that can assist in verifying a given claim. The content corresponding to these URLs is obtained and then fed into an NLI model to verify the claim.
The paper evaluates few-shot prompting strategies for obtaining the URL for LLM-Cite with a variety of LLMs. Experiments show that the approach is competitive and sometimes does even better than Google Search at identifying the relevant Wikipedia documents that can verify the claim. Further, given the high cost of Google Search, the approach is projected as a considerably cheaper alternative for identifying the URLs.

**Strengths:**

1) The approach shows competitive performance on a variety of benchmarks, although performance is considerably lower on the more recent FreshQA dataset.

2) The main strength of LLM-Cite is the cost savings compared to Google Search

**Weaknesses:**

1) My main concern is the novelty of the paper. The primary contribution is to show that the LLM can identify the relevant Wikipedia URL given a claim. However, this has already been demonstrated in LLM-URL [1], which considers question-answering tasks in a very similar setting, and is also missing in the related work.

2) Further, the paper considers the narrow setting of generating only Wikipedia URLs. Wikipedia URLs follow an entity-based template of https://en.wikipedia.org/wiki/<entity_name> which simplifies the task of generating relevant URLs to mainly just identifying the relevant entity in the claim to be verified. The authors should investigate how well this approach would work for tasks that require generating URLs from other websites, such as generating paper IDs or Arxiv URLs to verify scientific claims or generating news article URLs for news question answering.

3)  While the authors show cost comparison, there is no analysis done on latency comparison. Since the approach requires sampling or getting multiple URLs from an LLM such as GPT-4o, I would expect Google Search to be much faster in comparison.

[1] Large Language Models are Built-in Autoregressive Search Engines; Ziems et al ACL 2023

**Questions:**

1) While Google Custom Search is expensive at 5$ per 1000 queries, there are cheaper alternatives like https://serper.dev which offer a pricing varying from 1$ per 1000 queries to 0.3$ per 1000 queries. Given this, the authors should reconsider the claims made about 45x cost improvements.

---

> ### Author Response · Authors · 2024-11-21
> **Response to Reviewer CMf6**
>
> We thank the reviewer for the insightful comments and feedback! We try to address some of the concerns which were raised:
>
> > My main concern is the novelty of the paper.
>
> Thanks for pointing out the relevant work (we were not aware of this work before)! We will make sure to cite it. Even though both our work and that paper rely on generating URL we do want to emphasize a few key differences: (a) we focus on the freshness aspect of the URLs, where we show that it can help verify fresh claims; in fact, we show that in a few cases the generated URL is beyond the knowledge cutoff of the LLM (e.g. Figure 1) ; (b) we focus on fact verification and evaluate on corresponding datasets and compare to relevant baselines (e.g. FactScore) whereas the goal of the other paper was the retrieval part in QA — thus our work shows the applicability of generating URLs for a different task too.
>
>
> > Wikipedia URLs follow an entity-based template of https://en.wikipedia.org/wiki/<entity_name>  which simplifies the task of generating relevant URLs to mainly just identifying the relevant entity in the claim to be verified.
>
> We have tried to address this concern in the general response.
>
> > While the authors show cost comparison, there is no analysis done on latency comparison.
>
> We also briefly discuss latency in the paper (line 518 - 521). The estimated latency for LLM-Cite is small enough that it can be used in a production system too. We expect the total latency to be in the same ballpark as Google Search + NLI.
>
> > While Google Custom Search is expensive at 5$ per 1000 queries, there are cheaper alternatives like https://serper.dev which offer a pricing varying from 1 per 1000 queries.
>
> We want to note that APIs like SerpAPI rely on scraping search results, which will be unintended use of Google Search and not compliant with enterprise requirements. Hence, we decided to use Google Custom Search for our experiments and for cost comparison.

---

> > ### Comment · Reviewer_CMf6 · 2024-11-28
> > **Response to Author Rebuttal**
> >
> > Thank you for responding to my concerns.  However, I'm still not convinced about the novelty of the paper (for instance, fact verification and question answering share a lot of task similarities, in fact, a lot of papers have previously used QA for fact verification), and hence I will be maintaining my score.

---

### Author Response · Authors · 2024-11-21
**General Response to Reviewers**

We thank all the reviewers for their time and feedback on our work!

One common concern which was raised by multiple reviewers is that we only use Wikipedia in our setup. Multiple reviewers also pointed out that the structure of Wikipedia URLs is very special, and only requires identifying important entities and the URL always has the structure ‘https://en.wikipedia.org/wiki/<entity_name>’. We want to note a few points here:


1. We argue that a fact-checking system based on Wikipedia-only is nevertheless still a very practically useful system. Wikipedia has a wealth of factual information across 6million+ articles (english), and is commonly used for referencing checking claims. For instance, Natural questions (https://aclanthology.org/Q19-1026.pdf) shows that real google search queries are often answerable via wikipedia (sec 3.1). Additionally, a lot of the existing literature on fact checking has resorted to Wikipedia as the trustworthy source e.g. FactScore (https://arxiv.org/abs/2305.14251), Vitamin C (https://aclanthology.org/2021.naacl-main.52/), WikiCheck (https://arxiv.org/abs/2109.00835), etc.

2. Wikipedia URLs have a lot more variety beyond just <entity name> style URLs. This includes, URLs with list structure (e.g. https://en.wikipedia.org/wiki/List%20of%20New%20Zealand%20national%20cricket%20captains), history of entities (e.g.  https://en.wikipedia.org/wiki/History%20of%20Krispy%20Kreme), etc. Our approach can successfully  generate such URLs (see Figure 1 for an example), and can also generate <entity name> style urls where the entity is not explicitly mentioned in the query (e.g., generate https://en.wikipedia.org/wiki/Demographics_of_the_world for the query ‘The current population of the world is more than 8 billion as of 2024’. Generating such URLS is non-trivial and it greatly improves the recall of our method. This diversity of URL structure in Wikipedia further justifies our choice of using Wikipedia as a testbed for our approach.


We are also happy to make it more explicit in the paper that we only use Wikipedia e.g. adding Wikipedia in the title of the paper, as well as making clear throughout the paper. We do however discuss currently in Section 6, how the method can be extended to other domains such as NYTimes (which has semantically meaningful URLs) and arxiv (where we can create a cached key-value lookup with meaningful keys like title).

---

### Meta-Review · Area_Chair_TYAf · 2024-12-20

**Metareview:**

The paper introduces LLM-Cite, a method to verify factual claims by leveraging LLM-generated URLs to retrieve evidence instead of relying on traditional retrieval systems or static knowledge bases. LLM-Cite uses an LLM to generate potential citation URLs that can be fetched on-the-fly, enabling evidence retrieval for verifying claims. Retrieved URL content is processed through an NLI model to assess entailment and verify the claim. LLM-Cite is significantly cheaper than traditional methods like Google Search, reducing costs by more than 45 times. The method is simple, easy to deploy, and does not require complex external retrieval processes. It avoids reliance on static, un-updated knowledge bases or parametric knowledge within LLMs, ensuring up-to-date evidence with proper attribution. LLM-Cite is validated on three datasets, including both fresh and non-fresh claims from human and model-generated sources. Experiments show that LLM-Cite is competitive or even outperforms Google Search in identifying relevant documents for claim verification.

**Identified Strengths**:
1. LLM-Cite’s cost savings compared to traditional Google Search-based retrieval methods is a significant strength, offering a more accessible alternative for fact verification.

2. The proposed method is neat, simple, and easy to implement, making it scalable and valuable for real-world applications.

3. The methodology has been validated on diverse datasets, including human-written, model-generated, and fresh claims, showcasing its versatility and applicability.

**Identified weaknesses that need to be addressed**:
1. The method is primarily evaluated on Wikipedia URLs, where the structure is highly regular and semantically meaningful, but this restricts its applicability to other web domains with less structured URLs, such as PolitiFact or general web sources.

2. Experiments are limited to Wikipedia-based datasets, raising concerns about the method's generalizability beyond this specific knowledge base.

3. The accuracy of the approach heavily relies on the NLI model used for claim verification. The rationale for choosing specific NLI models (e.g., Gemini-1.5-Flash) is not discussed adequately.

4. Lack of validation on larger, well-established datasets like FEVER, HOVer, or FEVEROUS, which would strengthen the evaluation.

**Additional Comments On Reviewer Discussion:**

The authors tried to answer the common concerns of all reviewers related to the generalizability of the method and the overreliance on the semantic structure of generated Wiki URLs. However, I do not feel that to be convincing enough, and agree with the reviewers in their judgment on these concerns.

---

### Decision · Program_Chairs · 2025-01-22

Reject